# Dynamics of Fatty Acid Composition in Lipids and Their Distinct Roles in Cardiometabolic Health

**DOI:** 10.3390/biom15050696

**Published:** 2025-05-10

**Authors:** Fiorenzo Toncan, Radha Raman Raj, Mi-Jeong Lee

**Affiliations:** Department of Human Nutrition, Food and Animal Sciences, University of Hawaii at Manoa, 1955 East West Road, Honolulu, HI 96822, USA; rrraj@hawaii.edu

**Keywords:** n-3 fatty acids, membrane biology, cell signaling, lipid mediators, obesity, cardiometabolic diseases

## Abstract

Obesity and cardiometabolic diseases (CMDs) have reached epidemic levels. Dysregulation of lipid metabolism is a risk factor for obesity and CMDs. Lipids are energy substrates, essential components of cell membranes, and signaling molecules. Fatty acids (FAs) are the major components of lipids and are classified based on carbon chain length and number, position, and stereochemistry of double bonds. They exert differential impacts on CMDs, such that saturated fat increases risks while very-long-chain n-3 FAs provide benefits. The functionalities of FAs, modulating membrane properties, acting as ligands for receptors, and serving as precursors for lipid mediators, are vital for insulin signaling, lipid metabolism, oxidative stress, and inflammatory response, collectively contributing to cardiometabolic health. This review examines recent advances in the characteristics and functional properties of different FAs in lipid structures, signaling pathways, and cellular metabolism to better understand the differential roles of different types of FAs in obesity and cardiometabolic health.

## 1. Introduction

Cardiometabolic diseases (CMDs) have been prevalent in the past two decades [1]. Studies have described different characteristics and traits of CMDs as having components such as coronary heart disease, stroke, type 2 diabetes, dyslipidemia, and hypertension [2], while others describe CMDs as cardiovascular diseases (CVDs), stroke, and diabetes [3]. The American Heart Association provides a detailed description of CMDs, which includes obesity, severe obesity, dyslipidemia, hypertension, prediabetes, diabetes, chronic kidney disease, nonalcoholic fatty liver disease, and metabolic syndrome [1]. Lifestyle factors associated with CMDs are poor diet, physical inactivity, smoking, alcohol consumption, and stress. A key contributing factor to CMD development is lipid metabolism [1]. Lipids, often used interchangeably with fats, are a heterogeneous group of water-insoluble compounds that are soluble in nonpolar organic solvents [4]. Lipids can be classified based on their physical properties, polarity, or structure. The LIPID MAPS classification system categorizes lipids into eight groups: fatty acyls, glycerolipids, glycerophospholipids, sphingolipids, saccharolipids, polyketides, sterol lipids, and prenol lipids [5]. New additions to the LIPID MAPS are classes such as estolide, fatty aldehydes, and fatty alcohols [6].

Functions of lipids include energy substrate and storage, components of cellular membranes, lipoproteins, and lipid droplets, and roles in cell signaling. Triacylglycerol (TAG), a subclass of glycerolipids, is the principal storage form of energy that is mainly stored in lipid droplets in adipocytes in mammals [7]. When systemic energy demands increase, TAG is hydrolyzed through lipolysis, and FAs and glycerol are released from adipocytes. The amphipathic nature of phospholipids (PLs) enables them to span membranes, forming the basis for cellular membrane structures [8] and maintaining membrane integrity, which is the ability of membranes to resist, repair, and respond to different cellular contexts [9]. Membrane PLs also function as FA reservoirs for the synthesis of a variety of lipid mediators [10,11]. PLs can be classified as phosphatidyl-choline (PC), -ethanolamine, -serine, -glycerol, and -inositol, and sphingolipids based on polar head groups [12]. The two major classes of sphingolipids are sphingomyelin and glycosphingolipids [8]. PC is the most abundant PL in most tissues, while neuronal tissues contain more phosphatidyl-ethanolamine than -choline. They also have higher content of sphingolipids and ether lipids than peripheral tissues [13]. Cholesterol is also another common lipid, consisting of four rings, a hydroxyl group, and a hydrocarbon tail that can interact with sphingolipids [14]. Cholesterol is important for membrane integrity, modulating ion transport and signaling proteins in the membrane, and is a precursor to steroid hormones and vitamin D [8].

FAs are the major components of most lipids, and their composition in TAG and PLs reflects dietary intake [15]. The biological significance of FAs is extensive; they serve as vital energy sources, contribute to cell membrane structure, act as signaling molecules, and facilitate the absorption of fat-soluble nutrients [16]. Depending on the types of FAs, they exert differential impacts on CMDs such that saturated and trans fats pose risks while long-chain poly-unsaturated FAs (LC-PUFAs), especially n-3 eicosapentaenoic acid (EPA) and docosahexaenoic acid (DHA), are considered to be beneficial [17].

The Mediterranean diet, characterized by its high levels of PUFAs as well as Monounsaturated fatty acids (MUFAs) and fiber, is associated with reduced risks of CMDs and continues to be a cornerstone of nutritional interventions and therapeutic strategies [18,19]. However, randomized controlled trials investigating the effects of n-3 PUFA supplementation on CMDs have yielded inconsistent results. Systematic reviews and meta-analyses have shown improvements in glycemic markers, including HbA1c, fasting insulin, and fasting glucose levels [20,21], while others report null or contradictory findings [22,23]. Evidence also suggests potential benefits of n-3 PUFAs in reducing CVD risks [24,25] and fatty liver diseases [26,27]. Subgroup analyses indicate that efficacy may depend on CMD subtypes, baseline nutritional and metabolic status, dosages, and durations of interventions. Overall, these findings reflect significant heterogeneity in responses and highlight the need for larger, rigorously controlled trials to determine the therapeutic potential of n-3 PUFAs in CMDs. Further, recent studies have shown better cardioprotective effects of EPA, administered as ethyl-EPA, than commixed regimens of EPA/DHA [28]. Future studies using different formulations, dosages, or combinations of EPA and or DHA in specific groups are also warranted.

Our review explores the links between types of FAs and obesity, metabolic dysfunction-associated steatotic liver disease (MASLD), diabetes, and CVD, focusing on traits related to these CMDs, such as insulin resistance (IR), dyslipidemia, inflammation, and cellular stress pathways. We will also examine recent advances in FA properties in membrane biology, signaling activities, and cellular metabolism that can explain the deleterious effects of saturated FAs and the beneficial impacts of EPA and DHA on cardiometabolic health. Potential mechanisms that can contribute to the differential effects of EPA and DHA are also discussed.

## 2. Role of FAs in Maintaining Membrane Structure and Integrity

### 2.1. FAs in Lipid Structures of Cell Membranes

FAs are carboxylic acids (-COOH) that have an aliphatic chain (-CH_2_) and a methyl end (-CH_3_). Depending on their carbon chain length, FAs can be short- (SCFA, 2–4 carbons), medium- (6–12 carbons), long- (LCFA, 14–20 carbons), or very-long-chained (≥22 carbons) [29] based on the position of the first double bond counted from the methyl end. Unsaturated FAs contain double bonds (C=C) and are classified as MUFAs or PUFAs [30]. PUFAs are further classified into n-3 (omega-3) and n-6 (omega-6) depending on the position of the last double bond counted from the methyl end. Alkenes of the C=C can exist as *cis* or *trans* isomers, and *trans* FAs (TFAs) contain functional groups on opposite sides of the double bond. The most common TFA is elaidic acid, which is a *trans* isomer of oleic acid (OA, C18:1).

FAs are obtained through dietary lipids or de novo lipogenesis (DNL), which mainly occurs in the liver and adipose tissues in adulthood [31]. While some FAs are generated from acetyl-CoA through DNL, it is not a major pathway in adult humans consuming dietary lipids and is thought to play more regulatory functions [31]. Palmitate (PA) (C16:0) from the DNL is further elongated and desaturated into stearic acid (SA, 18:0) and oleic acid (OA, C18:1). However, humans do not contain delta-12 and delta-15 desaturases and need to consume two essential FAs, n-6 linoleic acid (LA, C18:2) and n-3 alpha-linolenic acid (ALA, C18:3), which are converted into arachidonic acid (ARA, C20:4, n-6), EPA (C20:5, n-3), and DHA (C22:6, n-3), respectively. ALA conversion into EPA and DHA in the human body is not efficient, and the general composition of FAs in PLs is attributed to dietary intake more than endogenous synthesis [31].

Membranes are composed of lipids (PLs and cholesterol) with embedded proteins and carbohydrate groups that are attached to them. PL structure is generally comprised of a glycerol backbone, FA tails, and a phosphate group. FAs have an amphipathic nature, enabling them to span membranes and to be the major structural component in PLs, and thus provide basic structure and functional properties in membranes [12]. FAs are attached to glycerophosphatides and sphingolipids through ester or ether bonds at the *sn-1* or *sn-2* position. Ether lipids have an alkyl chain attached to the *sn-1* position by an ether bond and constitute approximately 20% of PLs in mammals, while an ester bond is typically at the *sn-2* position [32]. SFAs are usually found at the *sn-1* position, and unsaturated FAs at the *sn-2* position [33]. After generation through the Kennedy pathway, PLs are remodeled through the Lands cycle, a de-acylation and re-acylation pathway, resulting in varied FA compositions [34]. Heterogeneities in FA composition between tissues have been shown and PA, the most common SFA, accounts for 20–30% of FAs in PLs with variable amounts of MUFAs (OA, ~19%), n-6 PUFAs (LA and ARA, ~25–60%), and n-3 PUFAs (ALA, EPA, docosapentaenoic acid [DPA, C22:5], and DHA, <30%) present in tissues [35].

### 2.2. FAs Influence Membrane Fluidity, Curvature, and Permeability

FA composition in membranes determines membrane permeability, morphology, and stability and has a significant impact on biological processes [36,37]. Membrane fluidity and curvature allow signaling molecules such as receptors, ligands, enzymes, and ion channels to function optimally [37]. Fluidity phases of membranes can be described as a solid-ordered phase, where lipids are in an aligned, rigid form, and a liquid-disordered phase, where acyl chains of lipids are more flexible and can move around [37]. The curvature of membranes entails the ability of cells to have flexible shapes, which are essential for physiological processes [14]. Additionally, PUFAs affect the formation and properties of lipid rafts, specialized domains within cell membranes, and play a key role in facilitating molecular interactions, compartmentalization, and signaling pathways that are essential for membrane signaling and cell traffic [38]. In the plasma membrane, lipid rafts are known to have a transient nature and include caveolae, which contain proteins involved in cell signaling [39].

Lipid model systems have demonstrated that membranes with PUFAs have defined elasticity and exhibit greater fluidity and flexibility, while TFAs and SFAs enable a straight chain structure that is more rigid [34] (Figure 1). *Cis* double bonds introduce bends or kinks in the FA chain, preventing closely packed FAs, creating a distortion in the bilayer, and increasing fluidity [33]. The disordering effect of PUFAs is also demonstrated to induce a thinner membrane compared to SFAs and cholesterol, which increase membrane thickness. The less thick membranes induced by a mixture of n-3 PUFA-PL can be attributed to their loose packing [40]. Membrane n-3 PUFAs interact with proteins involved in sterol transport, maintaining cholesterol distribution, PL packing, and permeability [41,42]. The degree of saturation and chain length of FAs also determines membrane permeability, and shorter chain FAs (C8 and C10) are known to destabilize bilayers through perturbing effects [43].

### 2.3. Impacts of FA-Induced Changes in Membrane Properties on CMD Risks

Membrane properties are implicated in the pathophysiology of metabolic disorders and CVDs [14], highlighting the importance of FA composition in membrane PLs. FA composition affects the mechanical properties of membranes, impairing receptor functions and signaling pathways. For instance, rigidity affects the Na^+^/K^+^-ATPase, leading to a reduction in the number of insulin receptors, whereas increasing PUFAs reverses this phenomenon [38]. Elevated levels of PLs enriched in SFAs and n-6 PUFAs compared to n-3 PUFAs are associated with compromised structural flexibility, impairment in insulin signaling, and increased IR [37,43]. Additionally, in the blood–brain barrier, lipid rafts composed of PUFAs enhanced insulin signaling [44], and DHA specifically ameliorated PA-induced impairment in insulin signaling in neuronal cells [45]. Palmgren et al., however, showed that increasing saturation of membrane PLs and rigidity did not affect insulin signaling activity in adipocytes, suggesting that adipocytes can withstand SFAs [46]. Therefore, more studies are needed to elucidate the underlying mechanisms that explain the potential cell-type-specific effects of SFAs on membrane functionality and signaling cascades. Additionally, the presence of DHA-PLs in non-raft areas induced movement of cholesterol and sphingomyelin into lipid rafts and increased raft size in macrophages, leading to reduced inflammation [47]. Whether other n-3 PUFAs have similar effects is not examined.

Studies in isolated model membranes have shown dissimilar effects of EPA and DHA in membranes, where EPA maintained intermolecular PL packing, while DHA had disordering effects and therefore caused cholesterol to self-aggregate and form cholesterol crystals [41]. DHA treatment decreased the ratio of cholesterol to PL, creating a more fluid membrane than EPA in aortic endothelial cells and platelets [28,48]. In conditions where glucose levels were elevated, cholesterol tended to aggregate, forming crystals, which increased reactive oxygen species (ROS) accumulation and oxidative stress [28]. EPA, being present in membranes, is known to more effectively inhibit ROS generation than DHA [41]. While these partially contribute to the better cardioprotective effects of EPA compared to the combined use of EPA/DHA [28], elucidating the mechanisms underlying the potential differential effects of EPA and DHA, as well as the potential counter-regulatory actions of DHA, remains an imminent area requiring further research.

## 3. Roles of FAs in Cell Signaling

FAs and their derivatives directly interact with membrane receptors and regulate cellular signaling pathways, affecting inflammation and metabolic activities. Depending on the types of FAs, different receptors and cellular signaling cascades are engaged, which partially explains the beneficial impacts of n-3 PUFAs as well as the deleterious effects of SFAs on CMDs.

### 3.1. Toll-like Receptors (TLRs)

SFAs have been shown to increase systemic inflammation and metabolic dysregulation through TLRs, pattern-recognition receptors that initiate responses to pathogens [49] (Figure 2). Upon ligand binding, TLRs recruit an adapter protein, MyD88, which activates IκB kinase and the phosphorylation and degradation of IκBα and β, inhibitors of nuclear factor-kappa B (NF-κB). This leads to translocation of NF-κB into the nucleus, where it induces the expression of genes involved in inflammatory responses [50]. TLR1–13 are present, and TLR4 is the most studied receptor for SFAs. SFAs, especially lauric acid (C12:0) and PA, are known to activate TLR4 signaling and inflammatory responses through stimulation of the NF-κB pathway and mitogen-activated protein kinases (MAPKs) in several cell types, including endothelial cells, macrophages, and cardiomyocytes [51,52,53]. PA activation of TLR4 requires MD-2, an accessory protein that is known to be crucial for the recruitment of MyD88 to TLRs in H9C2 cardiomyocytes, and deletion of MD-2 provides protection against PA-induced myocardial injuries [54]. Additionally, PA stimulates the NLRP3 inflammasome in hepatic stellate cells through the TLR4-NF-κB pathway, exacerbating the development of hepatic steatosis to fibrosis [55].

In addition to TLR4, SFAs have been shown to directly activate TLR2 and increase inflammatory signaling activities in monocytes and macrophages [56]. There is also a potential crosstalk between TLR2 and TLR4, where silencing of one may result in the upregulation of the other, thereby activating proinflammatory signaling through the NF-κB pathway in response to PA [51].

Aside from the canonical NF-κB pathway, studies have elucidated that TLR4 activation by PA can also upregulate *IL-6* expression via Kruppel-like factors (KLFs), a class of transcription factors with zinc finger structures [57]. It was previously established that KLF6 interacted with p65 NF-κB in the nucleus to trigger tumor necrosis factor-alpha (*TNF-α*) and *IL-1β* expression [58]. The same group observed that PA increased KLF7 by activating TLR4, and KLF7 directly binds to the *IL-6* promoter region, indicating a role of KLF7 in PA-mediated proinflammatory responses [57].

In addition to inflammatory responses, PA-TLR4 is known to affect many other cellular functions that are implicated in CMDs. PA-TLR4 activation led to impairment in cold-induced browning of white fat and thermogenic activation by inducing ER stress [59]. PA-TLR4 stimulation of ER stress was also involved in endothelial dysfunction and impairment in insulin-stimulated vasodilation [60]. PA also induced oxidative stress and apoptosis in vascular smooth muscle cells via TLR4-mediated induction of caspases and repression of p53 [61].

While numerous studies showed that SFAs activate TLR4 and 2, Lancaster et al. showed that TLR4 is not a receptor for SFAs [62]. Their study showed that TLR4-dependent alterations in cellular lipid metabolism indirectly contributed to the SFA-stimulated inflammatory signaling activities [62]. PUFA-mediated inhibition of TLR4 or antagonism of PA activation of TLR4 [63] may also involve alterations in lipid metabolism rather than their interference at the level of TLR4. Consistent with this hypothesis, EPA blocked PA-mediated induction of long-chain acyl-CoA synthetase 1 and inflammation in human THP-1 macrophages [64]. Whether other PUFAs, including ARA, DPA, and DHA, exhibit similar actions is undetermined.

### 3.2. Free Fatty Acid Receptors (FFARs)

FAs act as ligands for FFAR1–4, members of the G-protein-coupled receptor (GPCR) family, and regulate diverse biological processes, including taste perception, secretion of incretins and insulin, inflammatory responses, adipogenesis, and cellular metabolism [29]. FFAR2 (GPR43) and FFAR3 (GPR41) are activated by SCFAs, while FFAR1 (GPR40) and FFAR4 (GPR120) are activated by medium-chain FAs and LCFAs (Figure 3). PUFAs have a higher affinity for FFAR1 and 4 than SFAs, and DHA exhibits the highest affinity for FFAR1 [65,66]. Activation of GPCRs leads to the dissociation of α subunits from the heterotrimeric G-protein complexes, Gα/β/γ, and initiation of downstream signaling activities, including intracellular calcium levels, cAMP levels, phospholipase C, and MAPKs, depending on the types of α subunits engaged [67]. FFAR2 and 4 are coupled with Gq, which increases intracellular levels of calcium or DAG via activation of phospholipase C, while FFAR3 is coupled with Gi, which leads to decreases in cAMP levels through suppression of adenylate cyclase activity. Depending on cellular context, FFAR2 is also coupled with Gi, and FFAR1 with Gs, Gi, or Gq. Furthermore, similar properties of FAs activate more than one FFAR, although affinities may differ. These indicate the complexities of FA-FFAR signaling. Here, we briefly discuss the roles of FFARs in traits related to cardiometabolic health, and for more detailed information, we refer to previous reviews [29].

FFAR1 is highly expressed in enteroendocrine cells, pancreatic β-cells, and immune cells. In enteroendocrine cells, FFAR1 activation by LCFAs stimulates secretion of incretins, gastric inhibitory polypeptide (GIP), glucagon-like protein 1 (GLP-1), and cholecystokinin, indicating its roles in the regulation of insulin secretion and energy intake [68,69]. LCFAs (PA, LA, and DHA) have been shown to promote insulin secretion via FFAR1 in several cell culture models of β-cells and animal studies [66,70,71]. However, whether this leads to obesity-induced hyperinsulinemia remains inconclusive [70,72]. FFAR1 is also reported to be involved in anti-inflammatory responses mediated by EPA [73].

FFAR2 and 3 are shown to be expressed in enteroendocrine cells, β-cells, adipocytes, and immune cells, and they are activated by SCFAs, mainly acetate (C2:0), propionate (C3:0), and butyrate (C4:0) produced from gut microbial fermentation of fibers. While FFAR2 exhibits higher affinity for propionate and butyrate, FFAR3 has higher affinity for valerate (C5:0) [74]. Studies have shown that both receptors are involved in the secretion of gut peptide hormones, including GIP, GLP-1, and peptide YY, and hence the regulation of insulin secretion, glucose homeostasis, and energy balance [75,76]. Upon absorption, SCFAs activate FFAR2 and 3 and promote insulin secretion from β-cells and immune modulation [75]. In contrast, Tang et al. showed that acetate activated FFAR2 and 3 and inhibited insulin secretion by coupling to Gi in β-cells [77]. Their study also showed that FFAR2 and 3 in β-cells, but not in intestinal cells, mediate acetate inhibition of insulin secretion. FFAR2 is also implicated in the promotion of adipogenesis and suppression of lipolysis [78,79], while FFAR3 is known to be involved in the production of leptin, an adipocyte-derived anorexigenic peptide hormone [80]. More studies are needed to elucidate the roles of FFAR2 and FFAR3 in various tissues and their contributions to CMDs.

FFAR4 is the most implicated receptor in the beneficial effects of n-3 PUFAs. FFAR4 is highly expressed in the intestine and immune cells. In the intestine, FFAR4 functions as a receptor for PUFAs, promoting GLP-1 secretion by increasing intracellular calcium levels and ERK/MAPK activity [65]. Oh et al. showed that n-3 PUFAs suppressed inflammation in macrophages via FFAR4, improving insulin sensitivity [81]. When FFAR4 was activated by DHA, it associated with β-arrestin2, and the complex was internalized. This led to the inhibition of proinflammatory signaling pathways, including NF-κB and c-Jun N-terminal kinase (JNK). FFAR4 was also involved in anti-inflammatory actions of endogenously generated n-3 PUFAs in vascular inflammation and suppression of the thrombus formation and hyperplasia of neointima [82].

FFAR4 is also expressed in brown and brown-like (beige) adipocytes, and its expression levels are further increased upon cold exposure and catecholamine treatment through the p38 MAPK and protein kinase A (PKA) pathways, an indication of its role in thermogenesis [83]. Accordingly, FFAR4 activation has been shown to increase thermogenic activity by enhancing their metabolic activities or promoting the differentiation of progenitors into brown and beige adipocytes [83,84]. Induction of FGF21 through the p38 MAPK and alterations in microRNA networks have been proposed to be involved in the FFAR4-mediated browning process [85].

As expected from its roles, FFAR4-deficiency increased obesity upon high-fat-diet feeding and resulted in IR and hepatic fat accumulation with impaired insulin signaling activity and increased inflammation in adipose tissues [84,85]. Additionally, a lack of FFAR4 signaling activity due to a genetic mutation is correlated with increased risk of obesity [86]. While these results clearly demonstrate a critical role of FFAR4 in the protective effects on obesity and CMDs, other studies showed that FFAR4 is not needed for the anti-inflammatory and insulin-sensitizing effects of n-3 PUFAs [87,88]. Furthermore, although EPA and DHA bind to FFAR4 with high affinity, other LC-PUFAs, including LA and ARA, are known to activate FFAR4 [81,89]; therefore, how they regulate inflammation and metabolic activities through FFAR4 needs to be elucidated.

### 3.3. Transient Receptor Potential Channel of the Vanilloid Type (TRPV)

FAs also interact with TRPVs, a member of the transient receptor family that increases calcium influx into cells and activation of different protein kinases, including PKC and MAPKs. TRPV1–4 are primarily expressed in the central and peripheral neurons and are involved in sensing pain and noxious substrates as well as body temperature regulation [90]. Transgenic mice lacking TRPV1 exhibit reduced thermogenic gene expression in adipose tissues and develop obesity and metabolic diseases upon high-fat-diet feeding [91]. Other TRPVs have also been implicated in obesity and CMDs. TRPV2 is abundantly expressed in brown fat, where it mediates β3-adrenergic receptor-mediated increases in calcium influx and induces thermogenesis (Figure 3) [92]. On the contrary, expression levels of TRPV4 are higher in white than brown fat, and its activation suppresses thermogenic signatures while increasing proinflammatory pathways [93].

n-3 PUFAs, especially DHA and EPA, more effectively stimulated TRPV1 than n-6 PUFAs [94]. Fish oil containing n-3 PUFAs is known to activate TRPV1 in the gut, which leads to stimulation of the sympathetic nervous system and induction of thermogenic genes in brown and white fat depots [95]. However, an LA metabolite from the gut microbiome, 10-oxo-12(Z)-octadecenoic acid, has been shown to increase catecholamine turnover and thermogenic gene expression in brown and white fat by stimulating TRPV1 [96]. Additionally, ARA and its metabolites have been shown to stimulate TRPV4 [97], and whether this leads to the inhibition of thermogenesis needs further investigation.

Most findings originate from studies using cell culture and animal models. Therefore, whether these results accurately explain the differential impacts of FAs on CMDs in humans requires further confirmation. Furthermore, most studies have tested the effects of individual FAs, although several have indicated potential interactions between different types of FAs. Given that cells and tissues express more than one FA receptor and are exposed to a variety of FAs present in different concentrations, the in vivo scenario is complex. It is plausible that while LA, ALA, and ARA alone may exhibit null or deleterious effects, they might antagonize SFA-mediated actions, potentially leading to beneficial impacts. Consequently, more studies are needed to understand the complex interplay between FAs and membrane receptors in CMDs in humans.

## 4. FAs as Precursors for Signaling Molecules

In addition to acting as ligands for membrane receptors, FAs are involved in the synthesis of a broad range of signaling molecules and lipid mediators. These lipid-derived signaling molecules play crucial roles in inflammatory responses, cellular stress, and metabolic pathways, contributing to the pathophysiology of CMDs.

### 4.1. PA Increases Synthesis of Ceramides and DAG

Excess PA increases the accumulation of ceramide and DAG, inducing signaling pathways linked to insulin signaling and cellular stress. Palmitoyl-CoA combines with serine and initiates the biosynthetic pathway that leads to ceramide formation [13]. Elevated ceramide levels are commonly found in individuals with obesity and T2D [98,99]. In-depth lipidomic analysis in the DIVAS and HOLBAEK studies showed that ceramides and DAG were linked to cardiometabolic risk factors such as hepatic steatosis, dyslipidemia, and IR [100,101]. In the HOLBAEK study, 9 of the 17 main plasma lipid classes associated with CMD risk factors were ceramides [100]. Additionally, dietary modifications to reduce CMD risk decreased ceramide and DAG levels [100,101].

Ceramides have been shown to reduce insulin-stimulated GLUT4 translocation and glycogen synthesis in human myoblasts [102] and adipocytes [103]. Ceramide decreased the expression of AKT but also activated PKC, which further reduced insulin-stimulated AKT [104]. Additionally, ceramides decreased the activity of glycogen synthase kinase 3β, a key substrate of AKT, in C2C12 myotubes and 3T3-L1 adipocytes [105]. Ceramides activate a proinflammatory JNK MAPK, which interferes with insulin signaling cascades [106].

PA increased DAG levels in C2C12 myotubes and 3T3-L1 adipocytes, and DAG levels were correlated with impairment in insulin-stimulated AKT phosphorylation [105]. Increased DAG content in the plasma membrane leads to activation of novel isoforms of PKC, which impairs insulin receptor activity in the liver and muscle [107]. These findings indicate that PA-induced DAG also contributes to IR by interfering with insulin signaling cascades [102].

Ceramides also induce ER stress and other cellular pathways linked to cell toxicity. PA increased ceramide synthesis and induced lipotoxicity in rat β-cells and human endothelial cells [108,109]. Additionally, PA-derived ceramide was shown to induce ferroptosis by upregulating intracellular Fe^2^+ and suppressing antioxidant glutathione peroxidase 4 expression [110]. The study also showed that JNK MAPK is a downstream mechanism that mediates ceramide-induced ferroptosis [110]. While further studies are needed, these studies indicate that SFAs, especially PA, increase the risks of CMDs by inducing the synthesis of ceramides and DAG, which induce IR, proinflammatory reactions, and cellular stress signaling pathways.

### 4.2. PUFAs as Precursors for Lipid Mediators

PUFAs (ARA, EPA, DPA, and DHA), mainly derived from PLA2-mediated hydrolysis of membrane PLs but also from dietary intake or cellular TAG hydrolysis, are metabolized via cyclooxygenase (COX), lipoxygenase (LOX), or cytochrome P450 (CYP450) pathways to produce a myriad of lipid mediators that regulate diverse biological processes including inflammatory and metabolic processes [111,112,113]. There are two main COX enzymes, COX-1 and COX-2, and ARA metabolism through the COX pathway produces prostanoids, including prostaglandins (PGs: PGD2, PGE2, and PGF2α) and thromboxanes (TXs) (Figure 4). EPA metabolism via the COX family produces hydroxyeicosapentaenoic acids (HEPEs), and DHA via COX produces hydroxydocosahexaenoic acids (HDHAs).

LOX enzymes, 5-LOX, 12-LOX, and 15-LOX, catalyze ARA metabolism into hydroperoxyeicosatetraenoic acids (HpETEs), which can be further reduced to hydroxyeicosatetraenoic acids (HETEs) and leukotrienes (LTs: LTA4 and LTB4) [114]. LTs are further metabolized to lipoxins (LXs). EPA metabolism via the LOX pathway produces hydroxyeicosapentaenoic acids (HEPEs), which produce specialized proresolving mediators (SPMs), including E-class resolvins (Rvs). DHA via the LOX pathway produces hydroxyperoxydocosahexaenoic acids (HpDHAs), HDHAs, and SPMs, including D-class resolvins (RvDs), maresins (MaRs), and protectins (PDs). DPA is also used for the synthesis of SPMs. The CYP450 family produces HETEs and epoxyeicosatrienoic acids (EETs) from ARA while producing HEPEs from EPA and HDHAs from DHA. It is unclear whether DPA is also metabolized through the CYP450 system [17].

#### 4.2.1. Roles of ARA Derivatives in CMDs

ARA-derived PGs, LTs, TXs, and HETEs have proinflammatory, vasoconstrictive, prothrombotic, and angiogenic properties, which are critical for the cardinal inflammatory responses [115]. They also induce oxidative stress and decrease metabolic activities, further contributing to the pathophysiology of CMDs. On the contrary, EETs and LXs have been shown to exert anti-inflammatory and proresolutive actions, reduce oxidative stress, and increase metabolic activities [116]. An imbalance between these mediators is hypothesized to be the basis for chronic inflammation [117]. Therefore, uncovering molecular mechanisms balancing lipid profiles from a proinflammatory toward an anti-inflammatory/proresolutive profile could lead to the development of novel therapeutics for CMDs. The deleterious actions of ARA-derivatives have been extensively studied over the years, and we refer to a recent review that covered the topic extensively [115]. We will briefly discuss ARA derivatives that have been shown to exert beneficial impacts on CMDs, focusing on their modulation of metabolic activities.

EETs exhibit a protective role by promoting anti-inflammatory signaling pathways while inhibiting inflammatory factors, including IL-1β and IL-6, in several cell types, including endothelial cells and cardiomyocytes [118,119]. All four EETs (5,6-, 8,9-, 11,12-, and 14,15-EET) have been shown to induce peroxisome proliferator-activated receptor-gamma (PPARγ) expression and mitigate TNF-α-induced IκBα degradation, leading to the suppression of vascular inflammation [120]. EETs have been shown to exert protective effects by promoting the expression of antiapoptotic proteins by acting on PPARγ, PKB, and MAPK pathways in cardiomyocytes [121]. EETs also have been shown to exert beneficial effects, improving insulin signaling pathways and enhancing AMP-activated protein kinase (AMPK) and PPARγ [122]. EET stimulation of AMPK and PPARγ suggests their potential role in the improvement of metabolic diseases through remodeling of adipose tissues.

LXA4 and LXB4 and their aspirin-induced isomers, 15-epi-LXA4 and 15-epi-LXB4, exhibit anti-inflammatory characteristics [117]. Insufficiency of LXA4 resulted in inflammation, endothelial dysfunction, and progressive cardiac problems [123,124]. LXA4 and LXB4 reduced C-reactive protein and other proinflammatory cytokines and decreased oxidative stress in several cell types, including adipocytes, macrophages, and endothelial cells [125]. LXs act through formyl peptide receptor 2, decreasing NF-κB and increasing nuclear factor erythroid 2–related factor 2 (NRF2), thus decreasing inflammation and oxidative stress [126]. Additionally, LXs act through BLT1, a LTB4 receptor, to suppress inflammation [127].

PGE2, PGF2α, and PGI2, produced from ARA metabolism through COXs and PG synthases, have been implicated in the development as well as metabolic activity of both brown and white adipocytes. PGE2 is known to be important for the browning of white fat in mice, and the EP4 receptor, which is coupled with Gs increasing adenylate cyclase and subsequent induction of cAMP levels, mediates PGE2 signaling [128,129]. On the contrary, PGE2 and PGF2α have been shown to mediate ARA-suppression of the browning of white adipocytes differentiated from human multipotent adipose-derived stem cells through the activation of ERK MAPK [130]. Therefore, more research is needed to clarify their roles in the thermogenic capacity of brown and white adipocytes.

PGI2 increased the remodeling of white into brown-like adipocytes as well as the differentiation of brown-like adipocytes in isolated adipose progenitors from mouse and human white fat depots [129,131]. PGI2 activation of IP-receptor, which increases cAMP levels, followed by stimulation of the PKA and PPARγ, is known to be involved in the browning of white adipocytes and brown-like adipocyte development [131].

ARA-derived lipoxins, 15-epi-LXA4, LXA4, and LXB4, are reported to be the most abundant SPMs in brown fat in mice, and their levels are reduced after the development of obesity [132]. Further, transgenic mice with enhanced production of LXs in adipocytes exhibit enhanced browning and thermogenesis, and protection against obesity and IR [133]. While more studies are needed, LXA4 may enhance metabolic activity by increasing the synthesis of bile that acts through the liver X receptor or by suppressing proinflammatory signaling in adipose tissues [133], providing beneficial effects on CMDs.

#### 4.2.2. Roles of n-3 PUFA Derivatives in CMDs

Amongst RvEs, RvE1 is the most implicated one in CMDs. In humans, levels of RvE1, but not RvE3, exhibited consistent inverse associations with adiposity markers [134]. Further, patients with atherosclerosis and hypertension had lower levels of serum RvE1 than controls, while LTB4 levels were higher [135,136]. RvE1 significantly reduced atherosclerotic plaque and inflammatory cell infiltration in cholesterol-diet-fed rabbits [137] and suppressed leukocyte rolling and platelet aggregation [138]. RvE1 also attenuated doxorubicin-induced senescence in endothelial cells and cardiac fibroblasts by interfering with proinflammatory signaling activities, including the canonical NF-κB and NLRP3 inflammasome [139,140]. RvE1 signaling through ChemR23 decreased inflammation and immune cell infiltration via AMPK, NRF2, and the canonical NF-κB pathway, ameliorating hypertension and vascular remodeling [136]. RvE1 has been shown to protect against doxorubicin- or lipopolysaccharide-induced cytotoxicity by reducing oxidative stress through the regulation of AKT/mTOR signaling activity in cardiomyocytes [141,142]. Therefore, RvE1 acts through multiple signaling pathways and provides protective effects on CMDs.

In addition to acting through its receptor, ChemR23, RvE1 is known to antagonize LTB4-stimulated NF-κB by acting through BLT1 in leukocytes [127,143]. Furthermore, RvE1, but neither RvE2 nor RvE3, enhanced NADPH oxidase-mediated ROS generation via LTB4 receptors in polymorphonuclear cells [144]. These findings suggest that an imbalance between LTB4 and RvE1 may lead to a failure in the resolution of vascular inflammation and chronic inflammation in atherosclerosis.

Obesity is associated with reduced levels of PD1 and its precursor, 17-HDHA, and high inflammation. Additionally, dietary supplementation of EPA/DHA restored their levels and reduced inflammation [145], indicating their roles in inflammation, a major pathological factor in obesity-associated metabolic diseases. Accordingly, injection of RvD1, MaR1, and 17-HDHA suppressed inflammation, reduced macrophage accumulation, crown-like structures, and the expression of proinflammatory cytokines, and enhanced insulin-stimulated AKT phosphorylation in adipose tissues in mouse models [145,146,147]. MaR1 also increased the expression of adiponectin, an insulin-sensitizing and anti-inflammatory adipokine [147]. In human visceral adipose tissues and macrophages, RvD1 suppressed proinflammatory cytokine expression by blocking signal transducer and activator of transcription (STAT) 1 and STAT3, and enhanced resolution by increasing *IL-10* expression through antagonizing p38 MAPK activity [148]. As expected from the suppression of inflammation, RvD1, MaR1, and 17-HDHA improved insulin sensitivity in mouse models [145,146,147]. Additionally, PD1 was observed to enhance macrophage phagocytic activity and regulate leukocyte infiltration [149].

RvD1 and D2 attenuated angiotensin II-triggered recruitment of macrophages and reduced plasma levels of cytokines, perivascular fibrosis, and hypertension [150,151]. Similarly, RvD1 also blocked doxorubicin-induced inflammation in cardiac tissues [152]. In MASLD mouse models, RvD1 reduced proinflammatory cytokines through the suppression of the STAT1 signaling pathway and liver fibrosis [153]. RvD1 also inhibited oxidative stress and ER stress in both cardiac tissues and the liver and prevented cytotoxicity [152,153]. These results support the protective effects of DHA and DHA derivatives, especially RvD1, on CMDs. In addition to generating these protective lipid mediators, DHA supplementation also blocked the conversion of ARA into its proinflammatory derivative, 5-HETE [154], which could also contribute to the beneficial effects of DHA.

Several EPA and DHA derivatives have been implicated in the enhancement of adipocyte metabolic activities and thermogenic capacity. Leiria et al. showed that 12-HEPE and 14-HDHA are secreted from brown fat and involved in the enhancement of glucose metabolism through the activation of insulin signaling pathways in skeletal muscle and brown fat in mouse models [155]. Additionally, plasma levels of 12-HEPE and 14-HDHA are negatively associated with adiposity and IR, while positively correlating with brown fat activity in humans [155]. 12-HEPE may play a role in white adipocyte biology, as it has been detected in white fat and its levels are induced by n-3 PUFA supplementation and caloric restriction [156]. Other SPMs, including MaR2 and 4S,14S-diHDHA, are detected in brown fat in association with brown fat dysfunction, such as aging and obesity [132], indicating their roles in thermogenesis and protection against metabolic diseases.

### 4.3. Other Lipid Mediators

Several LA derivatives, 12,13-dihydroxy-9Z-octadecenoic acid, 9,10-epoxyoctadecenoic acid, and 9,10-dihydroxyoctadecenoic acid, and conjugated linoleic acids, have been shown to ameliorate metabolic health by enhancing thermogenic activities through activation of PPARγ and GPCRs [157,158,159]. Additionally, endocannabinoids produced from ARA regulate energy homeostasis and adiposity through modulation of energy intake and thermogenic capacity [160].

Studies have shown that branched fatty acid esters of hydroxy fatty acids (FAHFAs), characterized by a branched ester linked to a fatty acid and hydroxy fatty acid, are important regulators of biological processes in CMDs. Unlike PA itself, 5- and 9-palmitic acid-hydroxy stearic acids (PAHSAs) have been reported to improve insulin secretion and sensitivity and provide beneficial impacts on CMDs [161,162]. While the underlying mechanisms are not clearly elucidated, PAHSAs are shown to activate FFAR1 and 4. Different isomers of various FAHFAs, including 5-, 9-, 10-, 12-, and 13-PAHSA and FAHFAs containing different types of FAs have been identified to play roles in insulin secretion and insulin sensitivity [163]. Additionally, studies suggest that depending on the isomer and positioning of branching and carboxylic groups, FAHFAs can be anti-inflammatory [162]. While more studies are needed, these findings suggest that FAHFAs could be used as an intervention strategy for CMDs.

While the impacts of lipid mediators derived from ARA, EPA, and DHA on CMDs are discussed individually, they can compete for the same enzymes, and studies investigating how the balance between deleterious and protective lipid mediators is regulated in the human body are warranted. Further, DPA is also used for the synthesis of various SPMs that are known to exhibit protective effects on inflammation and CVDs [164], adding more complexity. Findings from recent deep-lipidomic studies indicate that lipid mediators can be used as biomarkers of CMD status as well as beneficial effects of dietary and lifestyle interventions [98,99,100,101]. Further, novel species of FA metabolites that exhibit therapeutic potential have been identified. Considering lipid mediators are often generated locally and temporally, advances in analytical techniques to identify novel lipid mediators and precisely measure their concentrations are needed.

## 5. Conclusions

Obesity and CMDs pose a significant health burden globally, and dysregulated lipid metabolism is a significant contributing factor to the diseases. Depending on the types of FAs consumed, their impacts on CMD risks vary, and incorporating dietary lipids containing n-3 PUFAs has been shown to effectively reduce risks associated with CMDs. Furthermore, purified n-3 PUFAs are used as a pharmacological approach to target specific aspects of CMDs. FAs, as components of PLs in membranes, ligands for various receptors, and precursors for lipid mediators, play crucial roles in the regulation of immune responses, oxidative stress, cellular signaling pathways, and metabolic activities. Chain length, the presence and position of double bonds in FAs, determine their functionality, which is related to their respective protective or deleterious impacts on cardiometabolic health.

In opposition to saturated and trans fats, PUFAs contribute to the maintenance of cellular membrane curvature and fluidity, enabling communication, ligand binding, and passage of molecules. Similar distinct effects are observed in many studies regarding their roles as ligands and precursors for lipid signaling molecules. While SFAs, especially PA, initiate an inflammatory response through TLRs and the canonical NF-κB pathways, PUFAs, especially EPA and DHA, engage with FFAR4 and exert anti-inflammatory actions and enhancement of adipose metabolic activities. PA increases ceramide and DAG accumulation, which interferes with insulin signaling and increases proinflammation and cellular stress pathways, further increasing the risks of CMDs. While ARA-derived PGs, TXs, and LTs are critical for cardinal inflammatory responses, several PGs have been implicated in the enhancement of thermogenic capacity. Additionally, ARA-derived EETs and LXs also provide benefits by improving vascular function, improving insulin sensitivity, and inhibiting inflammation. EPA-, DHA-, and DPA-derived SPMs not only provide protective effects but also antagonize the deleterious actions of SFAs and ARAs on cardiometabolic health. Therefore, elucidating the balance between signaling activities mediated by various FAs and their derivatives in cellular contexts is critical for the development of therapeutics for CMDs. Furthermore, most results are derived from pre-clinical models, requiring more studies to translate these findings to human health.

Both n-3 PUFAs, EPA and DHA, exhibit protective effects, yet emerging evidence suggests that EPA may offer specific advantages in reducing CVD risks, which is at least in part explained by its distinctive roles in membrane fluidity and stability. More studies investigating molecular and cellular mechanisms of their actions, optimal dosages, balances, and forms of EPA and DHA tailored to specific patients, long-term effects and safety of their uses, and their impacts on subpopulations of CMDs, including obesity, IR, and diabetes, MASLD, and CVDs, are needed.

## Figures and Tables

**Figure 1 biomolecules-15-00696-f001:**
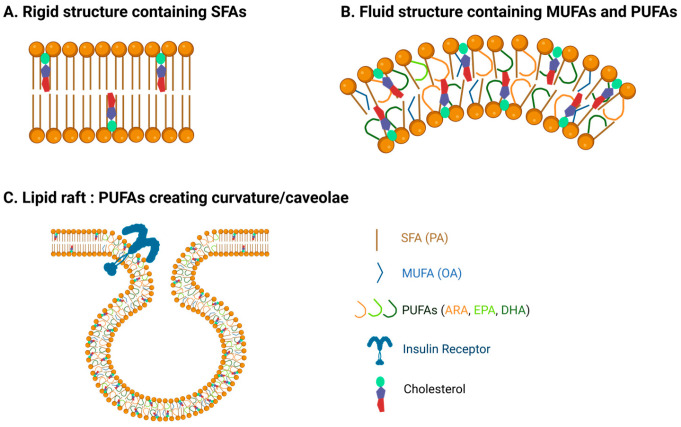
Fatty acids affect membrane properties. (**A**) Membranes with more saturated fatty acids exhibit more rigid, tightly packed, and thicker properties. (**B**) Poly-unsaturated fatty acids (PUFAs) increase membrane fluidity and curvature. (**C**) The presence of PUFAs in caveolae enables stability, curvature, and fluidity, facilitating the concentration and function of proteins such as insulin receptors.

**Figure 2 biomolecules-15-00696-f002:**
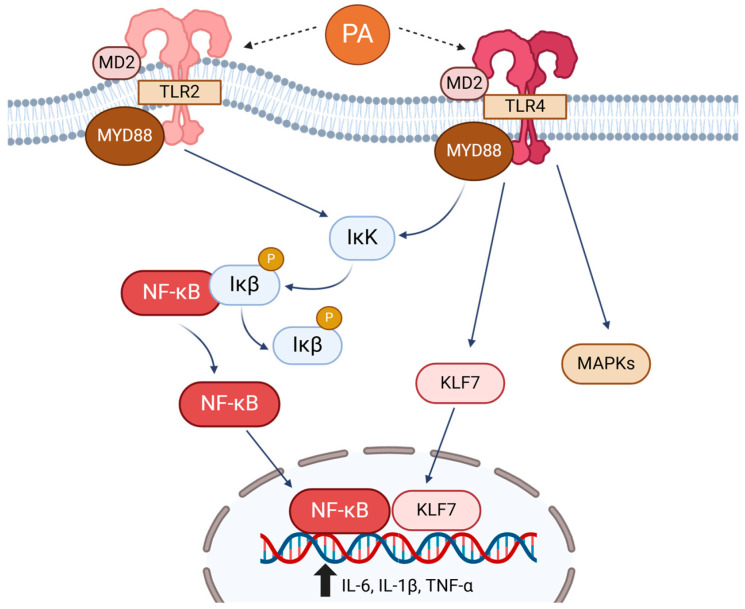
Saturated fatty acids (SFAs), especially palmitic acid (PA), activate Toll-like receptor (TLR) 4 and TLR2, triggering the canonical NF-κB pathway and proinflammatory responses. PA via TLR4 also activates Kruppel-like factor 7 (KLF7) and mitogen-activated protein kinases (MAPKs), increasing proinflammatory responses.

**Figure 3 biomolecules-15-00696-f003:**
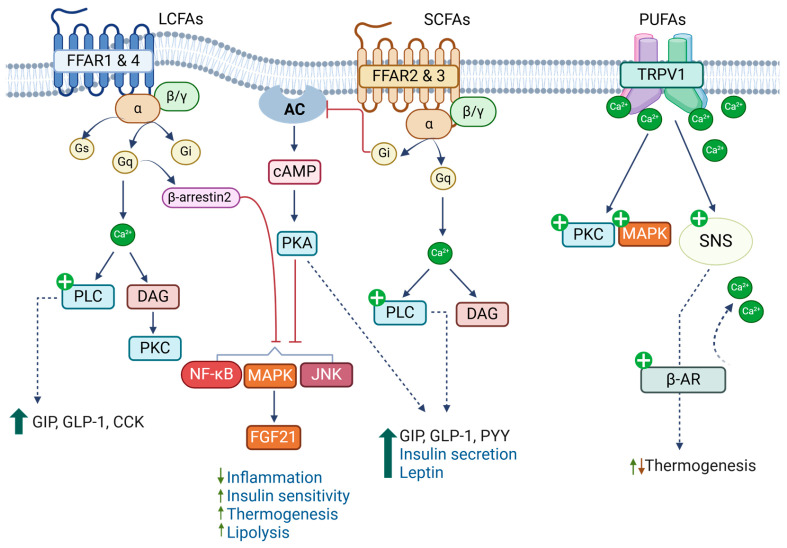
Fatty acids regulate biological processes by acting as ligands for membrane receptors. Long-chain fatty acids (LCFAs) activate FFAR1 and FFAR4, regulating incretin and insulin secretion, inhibiting inflammatory responses, and improving thermogenesis. Short-chain fatty acids (SCFAs) activate FFAR2 and FFAR3 and regulate secretion of gut-derived peptide hormones, insulin, and leptin. Poly-unsaturated fatty acids (PUFAs), especially n-3 PUFAs, activate the membrane potential channel of the vanilloid type 1 (TRPV1) and regulate thermogenesis.

**Figure 4 biomolecules-15-00696-f004:**
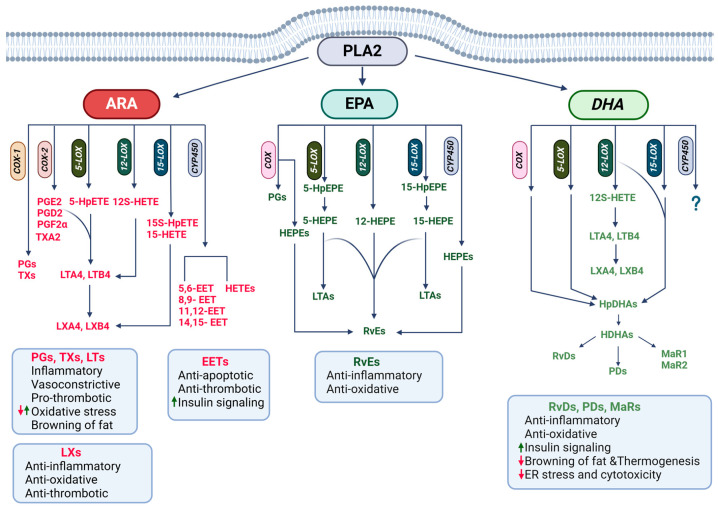
Lipid mediator regulation of biological processes involved in cardiometabolic diseases. Arachidonic (ARA), eicosapentaenoic (EPA), and docosahexaenoic (DHA) acids derived from phospholipase A2 (PLAD)-mediated hydrolysis of membrane phospholipids are metabolized via cyclooxygenase (COX), lipoxygenase (LOX), and cytochrome P450 (CYP45) pathways to produce bioactive lipid mediators that regulate inflammation, metabolic pathways, and stress responses.

## Data Availability

Not applicable.

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
