# Peer review of "Dynamics of Fatty Acid Composition in Lipids and Their Distinct Roles in Cardiometabolic Health"

_biomolecules, 2025, doi:10.3390/biom15050696_

Round 1
Reviewer 1 Report
Comments and Suggestions for Authors
Fiorenzo, Radha Raman and Mi-Jeong have done a commendable work in summarizing the decades of seminal work by many investigators in the field of lipid/adipose metabolism. This is a very balanced review that highlights how certain fatty acids reduce the risk of CMD while others promote CMD development. Job well done.
Author Response
Thank you for reading our manuscript.
Reviewer 2 Report
Comments and Suggestions for Authors
It is anticipated that as a part of special issue this review would provide an in-depth and critical assessment of the current knowledge identifying key gaps and constructive future directions to the readers. The comments provided by the reviewer is meant to be constructive and are as follows:
The title needs refining and needs clarity. Not sure what “Dynamics of fatty acid composition in Lipids ..” means? For instance, which lipid types and where?
A possible title could be “Effect of cellular fatty acid composition on the dynamics of molecular pathways, relevant to cardiometabolic diseases: A review focusing on preclinical models”.
Most of the content of this review is based on molecular mechanisms which is highly appropriate to the mandate of the journal, The review is written well, however, will need further editing and refining. For example, classification of omega 3 or 6 fatty acids based on the location of double bond, is incorrect (line 77-78), number of statements made throughout the text needing clarification. Reference list needs attention with respect to completeness etc, such as reference #25, 35, 89 etc.
It is a comprehensive general review focusing on the role of fatty acids in modulating membrane structure and function (membrane receptors, molecular pathways) and metabolites of long chain omega-3 and 6 fatty acids including their impact on cell function (molecular pathways). These pathways are integral to many cellular functions from energy metabolism to cell growth and differentiation, inflammation etc. These pathways are not specific to CMDs and have been reviewed previously as cited by the authors and are present in the literature related to other chronic diseases one important example is “cancer”. Moreover, several other nutrients seem to also affect the specific molecular pathways, noted in this review.
Clarification is needed regarding the purpose of this review as highlighted by the authors. The reviewer found the suggested purpose to be confusing. The authors suggested that the purpose of this review is to explore-1) the links between types of FAs and obesity, metabolic dysfunction such as associated steatotic liver disease, diabetes and CVD, focusing on traits related to these CMDs, such as insulin resistance (IR), dyslipidemia, inflammation, and cellular stress pathways; and 2) examine recent advances in FA properties and metabolism, with a focus on the protective effects of EPA and DHA on cardiometabolic health. Should the title of the review include this focus??
Why focus on the protective effect exclusively? Are there deleterious effect of long chain PUFA in the membrane on the susceptibility to oxidative damage as well as disturbed signaling pathways?
In the abstract, focus seems to be related to LC omega-3 fatty acids, however a distinct section on how the dynamics of various molecular pathways differ when LC omega-3 replaces LC omega-6 or saturated fatty acids, is missing. Are there studies in which a comparison among groups in which treatment groups included saturated fatty acids, LC omega-6 and LC omega-3 fatty acids on the molecular pathways described in the review? The review appears to be general in nature. The content of the review does not fully support the content of the abstract.
The review will benefit by being more specific and by removing vague statements. For example, population-based studies do not support the fact that LC omega-3 is protective. To the reviewer’s knowledge the protective effects of EPA and DHA on CMDs remain uncertain (lines 63-64), line 162-163- Be more specific and provide examples as to what type of studies may be needed.
This review is heavily dedicated to saturated fatty acids or scattered evidence from different fatty acids. In keeping with the abstract, it is imperative that additional discussion pertaining to omega-3 or omega-6 fatty acids should be provided. For example, see section 3.1 Toll-like receptors.
Throughout the review several organ types are mentioned. Any idea which organ and cell types or cellular organelles are more critical in the development of CMDs in responding to type of fatty acids and Which membrane types are discussed?
For example, mitochondria as organelles in cardiomyocytes or many other tissues could play a major role in the initiation of symptoms associated with CMDs.
Do unsaturated as opposed to saturated fatty acids affect mitochondrial health and dynamics?
An additional section related to a critical analysis of the findings from pre-clinical models and the relevance of numerous molecular mediators and mechanisms in real world situations will be valuable. For example, we know that CMDs involve multistep and multifactorial processes and may involve failure of physiological homeostasis in multiple organs. Whether, during the development of CMDs there is a process or organ type more profoundly affected by the type and quantity of fatty acids ingested initiate the traits of CMDs, needs to be highlighted.
Is the quantity of fatty acids, regardless of the type (quality), critical in CMDs or in altering molecular dynamics?
As the authors have pointed out, CMDs involve a range of physiological abnormalities and therefore, the review encompasses findings from various models representing different physiological states and organ types. The novel contribution of this review must be highlighted.
The review will benefit by having a section on why LC omega-3 should be investigated in future studies, with supporting evidence. Why not PUFA as opposed to saturated fatty acids?
Briefly, in the reviewer’s assessment this document will benefit from the following:
- The title needs to represent the content of the manuscript, and any ambiguity should be removed.
- The entire review will benefit from a focused approach and by highlighting its novel contributions to the field of CMDs.
- A balanced view and critical assessment of the existing knowledge be provided.
- It should be acknowledged that most of the information presented in the document is derived from pre-clinical models and that the role of LC omega-3 fatty acids in human studies remain debatable.
Author Response
We truly appreciate your thorough reading of our manuscript and providing constructive feedback. We addressed majority of your and other reviewers’ comments in the revised manuscript, marked in red.
The title needs to represent the content of the manuscript, and any ambiguity should be removed.
Response to comment: We would like to thank the reviewer for the thorough evaluation of our manuscript and insightful comments. We have revised the manuscript title to better reflect the main scope of the review. The new title emphasizes the role of various fatty acids in cardiometabolic diseases.
The entire review will benefit from a focused approach and by highlighting its novel contributions to the field of CMDs.
Response to comment: Thank you, we tried to address this in the revision.
A balanced view and critical assessment of the existing knowledge be provided.
Response to comment: We revised multiple sections to provide a more balanced perspective on the current body of literature. This includes a more critical evaluation of both the strengths and limitations of the existing data and highlighting areas where further research is needed.
It should be acknowledged that most of the information presented in the document is derived from pre-clinical models and that the role of LC omega-3 fatty acids in human studies remains debatable.
Response to comment: We have now stated that much of the evidence presented is derived from pre-clinical studies and have emphasized the ongoing debate surrounding the efficacy of long-chain omega-3 fatty acids on cardiometabolic diseases.
Fiorenzo Toncan
Reviewer 3 Report
Comments and Suggestions for Authors
Metabolic disorders constitute a major problem of public health that is associated with increased risk of mortality and poor quality of life. Poor dietary and habitual lifestyle are considered critical issues that worsen the prognosis of patients suffering from low-grade inflammatory diseases (i.e., glucose and lipid metabolic disorders). In this aspect, the present review aims to overview the various dietary lipids used to manage the prevailing epidemic of cardiometabolic health. This study reviewed the dynamics of fatty acid composition in animal and clinical subjects. However, such results and conclusions were well-known information. Therefore, I still have some questions raised for the manuscript.
- This review aims to highlight recent possible contributions to alleviating various dietary lipids in managing cardiometabolic health in metabolic diseases based on the rising prevalence of obesity. The possibility of nutritional status and lipid contributors in disease progression and management should be focused. According to the currently available clinical data, the nutritional status of patients seems to play a very important role in the development and progression of obesity, metabolic, and even neurodegenerative diseases. A recommended nutritional approach or nutrition intervention may be necessary to reveal or even alleviate the neurodegenerative disease and obesity patients monitor the development of their disease.
- This is a brief review article on cardiometabolic disease and dysfunctional symptoms in metabolic diseases, especially in T2D and stroke. The mentioned dynamic status of fatty acids in different diseases should exclude the dietary factors (lipids and carbohydrates); such information is valuable for clinics. From the disease prevention perspective, some updated potential biomarkers should be discussed. Overall, this is a well-organized brief review article.
Author Response
Thank you for your careful reading and suggestions. We revised manuscript based on the reviewers comment.
Response to comment 1: We appreciate the reviewer’s insightful comment. We briefly discussed the role of nutritional status and results from clinical trials of n-3 fat in the progression and management of cardiometabolic diseases. While we agree that the reviewer’s comment regarding nutritional approach for neurodegenerative disease, this is beyond scope of our manuscript.
Response to comment 2: We now addressed potential lipid derived biomarkers in the revision.
Fiorenzo Toncan